# A Novel Registration Method for a Mixed Reality Navigation System Based on a Laser Crosshair Simulator: A Technical Note

**DOI:** 10.3390/bioengineering10111290

**Published:** 2023-11-07

**Authors:** Ziyu Qi, Miriam H. A. Bopp, Christopher Nimsky, Xiaolei Chen, Xinghua Xu, Qun Wang, Zhichao Gan, Shiyu Zhang, Jingyue Wang, Haitao Jin, Jiashu Zhang

**Affiliations:** 1Department of Neurosurgery, First Medical Center of Chinese PLA General Hospital, Beijing 100853, China; chxlei@mail.sysu.edu.cn (X.C.); dr_xxh@126.com (X.X.); wfwangqun@163.com (Q.W.); 15620946575@163.com (Z.G.); sy983246393@gmail.com (S.Z.); richardwang9410@163.com (J.W.); 18031275391@163.com (H.J.); 2Department of Neurosurgery, University of Marburg, Baldingerstrasse, 35043 Marburg, Germany; christopher.nimsky@uk-gm.de; 3Center for Mind, Brain and Behavior (CMBB), 35043 Marburg, Germany; 4Medical School of Chinese PLA, Beijing 100853, China; 5NCO School, Army Medical University, Shijiazhuang 050081, China

**Keywords:** mixed reality navigation, augmented reality, neurosurgical interventions, preoperative planning, registration method, laser, crosshair simulator, accuracy, target registration error, head phantom

## Abstract

Mixed Reality Navigation (MRN) is pivotal in augmented reality-assisted intelligent neurosurgical interventions. However, existing MRN registration methods face challenges in concurrently achieving low user dependency, high accuracy, and clinical applicability. This study proposes and evaluates a novel registration method based on a laser crosshair simulator, evaluating its feasibility and accuracy. A novel registration method employing a laser crosshair simulator was introduced, designed to replicate the scanner frame’s position on the patient. The system autonomously calculates the transformation, mapping coordinates from the tracking space to the reference image space. A mathematical model and workflow for registration were designed, and a Universal Windows Platform (UWP) application was developed on HoloLens-2. Finally, a head phantom was used to measure the system’s target registration error (TRE). The proposed method was successfully implemented, obviating the need for user interactions with virtual objects during the registration process. Regarding accuracy, the average deviation was 3.7 ± 1.7 mm. This method shows encouraging results in efficiency and intuitiveness and marks a valuable advancement in low-cost, easy-to-use MRN systems. The potential for enhancing accuracy and adaptability in intervention procedures positions this approach as promising for improving surgical outcomes.

## 1. Introduction

In recent years, commercial navigation systems have become a critical paradigm in neurosurgery [1,2,3], enabling neurosurgeons to match preoperative images to the patient space, track the position of surgical instruments within the patient’s body, control the surgical scope, and protect eloquent anatomical structures [4,5]. Pointer-based navigation, an early navigation standard, requires specialized navigation pointers as definable structures for tracking. In most instances, neurosurgeons need to switch surgical instruments with the pointer when navigation support is needed, thus interrupting surgical operations. In addition, neurosurgeons often repeatedly shift their attention between the navigation pointer in the surgical area and the nearby navigation monitor, causing distractions and fatigue [6,7,8,9,10]. Therefore, this paradigm has poor ergonomics.

Augmented Reality (AR) technology visualizes virtual information within real-world scenes and virtual content and represents a significant technological breakthrough in neuro-navigation, specifically microscope-based navigation [11,12,13,14,15]. This navigation paradigm employs the geometric optical focus of the microscope as a virtual pointer tool and overlays two-dimensional (2D) or three-dimensional (3D) virtual anatomical structures from the image space to the actual surgical area; i.e., the microscope’s focal plane [3,13,14,16,17]. The clinical benefits of integrating AR technology with navigation systems have been extensively validated: it enhances surgeons’ comfort, reduces attention shifts, and heightens physicians’ intuitive understanding of anatomical structures, which is especially beneficial for less experienced surgeons [3,13,15,16,17]. Nonetheless, microscope-based navigation comprises substantial hardware components, including infrared tracking cameras, navigation workstations, integrated microscopes, and accessories, often associated with high cost. Moreover, procuring AR and navigation support requires extra preoperative and intraoperative procedural time. It necessitates that the entire team—surgeons, operating room staff, and anesthetists—be sufficiently familiar with this paradigm [3,15,16,17].

The substantial, expensive hardware systems and complex workflows challenge the widespread adoption of microscope-based navigation. Hence, there has been increasing interest in smaller, low-cost, and easy-to-use alternatives for AR-guided interventions, such as projector-based [18,19], smartphone-based [20,21,22,23,24], tablet-based [22,25,26,27], and head-mounted display (HMD)-based AR [7,8,10,28]. These AR paradigms are widely used as “low-cost navigation” in underdeveloped regions or healthcare facilities with limited resources. However, projectors suffer from perspective effects and geometric distortion, unable to provide depth information for virtual objects [18,19]. Once the user deviates from the projector’s projecting direction, the parallax between the virtual model and the actual anatomical structure confuses the AR experience, making projector AR more suitable for the preoperative localization of superficial cranial lesions [18,19,29]. Smartphones and tablets have sufficient computational resources for 3D rendering, model smoothing, and ambient light simulation, enhancing users’ sense of depth and realism when perceiving virtual objects [20,21,25,26]. Nonetheless, given the limited screen size, they provide a restricted field of view. Moreover, user interactions with them depend on mouse or keyboard input, which is incompatible with sterile surgical requirements [20,21,25,26]. Considering these limitations, HMD-based AR may be a more valuable approach. Its advantages include hands-free and portable operation (although weight and volume still need improvements), a larger field of view, and a more immersive visualization environment. HMDs can be categorized into video see-through and optical see-through paradigms based on whether users can directly see the real world [30]. The latter offers a more realistic and immersive user experience compared to the former, making optical see-through HMD-based AR the most promising and appealing low-cost AR paradigm.

Mixed Reality (MR) technology, integrating real-world scenes with virtual space, provides a new environment for users to interact with virtual content in the real world. Although a clear distinction between MR and AR has not been universally agreed upon, MR is considered a technology between virtual reality (VR) and AR in many publications [6,7,8,9,31,32,33,34,35,36]. This view has some validity because MR technology typically integrates optical and physical sensors that digitalize real-world information into the virtual world, making this information computable and interactive rather than merely superimposing virtual and actual elements, as in AR. Therefore, MR is the term used in this paper. The Microsoft Corporation launched the world’s first commercial stand-alone wireless MR HMD, HoloLens, in 2016. The advanced Synchronous Localization and Mapping (SLAM) technology allows HoloLens to understand its position relative to the actual environment and stably integrate 3D holograms into the real world. This technology lets virtual content be directly displayed in the user’s visual field. In 2017, Incekara et al. implemented the first proof-of-concept MR navigation (MRN) system in clinical settings [6]. Based on preoperative magnetic resonance imaging (MRI) data, they created 3D holograms of the patient’s head and tumor. They manually aligned the holograms to the patient’s head to localize the intracranial tumor. At that time, research mainly focused on the practical benefits of MRN to patients and neurosurgeons and its applicability in various neurosurgical intervention procedures [6,7,9,10,37].

Over the subsequent years, it was unsurprising that research publications on MRN in neurosurgery exponentially increased [30,38,39,40,41,42,43]. MRN offers neurosurgeons an immersive 3D neuroanatomical interactive model at a lower cost and with an easier-to-use system configuration than conventional pointer-based and microscope-based navigation systems [6,7,8,9,10]. However, as studies delved deeper, issues arising from this new technology became apparent, such as a highly user-dependent holographic registration process and relatively higher spatial localization errors [6,7,8,10,31,44,45,46,47]. Therefore, it is generally acknowledged that, despite its advantages, MRN is not yet capable of replacing conventional navigation systems.

The benefits and safe application of MRN for neurosurgical interventions fundamentally depend on the precision and accuracy with which virtual objects blend with real-world scenarios. Inaccurate or ambiguous virtual content may only offer physicians a deceptive sense of security, potentially posing patient risks. According to Peter et al. [34], the clinical accuracy of a system should constitute a combination of registration precision (objective) and user perception accuracy (subjective). User perception accuracy, being dependent on individual visual-spatial abilities and prior experience, is beyond the scope of technological improvement. Nonetheless, registration accuracy is primarily influenced by the initial registration accuracy, suggesting that improvements in the technical aspects of the initial registration process could enhance the overall accuracy. Beyond the fully manual registration methods reported in early MRN prototypes [7,8,32,48], several other registration methods similar to those in commercial navigation systems have proven technically feasible. These include point matching of anatomical landmarks, cranial fixations, affixed skin markers [9,10,35,36,44,49,50,51,52,53], and surface matching [33,54,55,56,57,58,59]. Nevertheless, due to marker shifts or skin shifts during imaging relative to patient positioning in the OR, coupled with users’ errors while acquiring points on the surface or from the markers with physical tools, the reported MRN registration methods are almost entirely user-dependent [46,55,56]. This has resulted in reported reference accuracies varying from 0–10 mm in clinical settings, which are not entirely acceptable for high-precision neurosurgery.

In summary, existing MRN registration methods have yet to concurrently fulfill the requirements of low user dependency, high accuracy, and clinical applicability. However, with the increasing demand for neuroimaging in multiple situations, compact and portable scanners (e.g., intraoperative Computed Tomography (CT) [1,60,61], cone beam CT (CBCT) [62,63,64], portable MRI [65,66,67]) have been widely deployed in ORs and treatment rooms, making the acquisition of 3D imaging no longer confined to radiology departments [1,3,60,61]. This development has given rise to an automatic registration paradigm. This paradigm reliably maps the acquired volumetric data (reference image) to the acquisition position by tracking the scanner, thereby ensuring accurate navigation registration [1,60,61]. This paradigm has been maturely applied in multimodal commercial navigation systems, demonstrating its potential to reduce user dependency and increase accuracy, which might be also applied in MRN, providing a promising approach to overcome current limitations of MRN.

Therefore, a novel MRN registration method was developed in this study using registration with a laser crosshair simulator by replicating the image acquisition position on the patient using laser crosshairs and automatically calculating a transformation of tracking space and reference image.

## 2. Materials and Methods

This section delivers an in-depth description of the design and calibration principles underlying the crosshair simulator, followed by an explanation of the mathematical model and operational process of the crosshair simulator-based registration. This section concludes with an experimental design for assessing the accuracy and feasibility of the proof-of-concept MRN system.

### 2.1. The Crosshair Simulator

#### 2.1.1. Concept and Design of the Crosshair Simulator

In commercial navigation systems, the most classic and practical solution for automatic registration is to equip the scanner with reference markers and use a tracking system during image acquisition to convert the scanned reference images into a common coordinate system, such as the world coordinate system [1,60]. Nonetheless, this approach is challenging to apply directly to MRN. On the one hand, the optical tracking components are integrated into the HMDs, requiring users to stay away from the scanner during scanning, potentially interrupting optical tracking (see Figure 1A). On the other hand, movements of large objects in the room, e.g., CT scanner translation, can introduce significant localization errors in HMD’s spatial mapping.

To address this issue, the concept of a crosshair simulator is proposed (see Figure 1B). The crosshair simulator aims to transfer scanning parameters between different spatiotemporal domains to determine the acquisition position of the reference images (see Figure 1B and Figure 2A), and essentially consists of a rack, two laser emitters, and an MR interface.

The rack is an “L” shaped basic frame for mounting other components such as the laser emitters. Users can flexibly adjust its position and orientation in space using a handle and then securely lock it in place with a mechanical arm (see Figure 2A).Two laser emitters (wavelength: 650 nm, power: 12 mW) are horizontally and vertically fixed on the arms of the rack. They project two sets of laser crosshairs inside the simulator, with the centerlines of the crosshairs located coplanar and perpendicular to each other. This configuration creates three orthogonal planes in space, forming the simulator coordinate system (SCS). When an object is exposed to the laser, it will receive a crosshair projection on its top and side surfaces, analogous to the positioning crosshairs observed in CT or MRI scanners (see Figure 2B).The MR interface consists of a stainless-steel panel (size: 6 cm × 6 cm) printed with a visually recognizable target image. It is firmly fixed on the rack. The MR interface establishes the relationship between the tracking and virtual spaces. Once the target images are detected and recognized by the HMD, the virtual space is initialized with the geometric center of the target image as the origin. This process is implemented using the Vuforia Software Development Kit (SDK) (Version 10.14, PTC, Inc., Boston, MA, USA).

#### 2.1.2. Coordinate Systems

The crosshair simulator includes three coordinate systems (see Figure 3):The simulator coordinate system (SCS) is defined from the laser crosshairs’ geometric-optic relationships.The reference image coordinate system (RICS), defined by the scanner during reference image acquisition, either in MRI or CT procedures. This coordinate system is established at the position where the gantry laser positioning lines are projected. In the case of MRI, the first “localizer scan” at the beginning of the scanning session establishes this coordinate system, while in CT procedures, a similar “scout scan” is used for the same purpose.The virtual coordinate system (VCS) is defined by recognition of the MR interface.

The transformation $T_{V}^{S}$ from the VCS to SCS reflects the fundamental purpose of the crosshair simulator, i.e., to establish a mapping relationship between the virtual space defined by the image target on the MR interface and the physical space defined by laser crosshairs. It can be calculated as Equation (1):(1)$${T_{V}^{S}=T}_{R}^{S}\cdot T_{V}^{R}$$

The transformation $T_{R}^{S}$ (intrinsic matrix) from RICS to SCS describes the inherent properties of the simulator for scanning parameters transmission. The simulator laser positioning is the inverse process of scanners utilizing laser positioning, as the scanner realizes the mapping of scanning parameters from physical to image space. In contrast, the crosshair simulator implements the inverse mapping. A calibrated $T_{R}^{S}$ ensures that the three reference planes of the scanned image are aligned with the positions of the three laser locating planes of the scanner, thereby coinciding with the mechanical center and axes of the scanner (see Figure 1B, Figure 3 and Figure 4A–C).

The transformation $T_{V}^{R}$ (extrinsic matrix) from the VCS to RICS defines the position and orientation of RICS in the VCS. With this transformation, the imported model in the virtual space can be mapped to the correct position in the RICS, enabling visualization through the HMD (see Figure 1B, Figure 3 and Figure 4D,E).

Among calibrated crosshair simulators of different specifications, their external matrices may vary due to the spatial relationship between the image target’s geometric center and the laser system’s center. Still, their intrinsic matrices must be identical, as the principle of scanning parameter transmission remains the same.

#### 2.1.3. Calibration

To ensure the reliability of the crosshair simulator, both the intrinsic and extrinsic matrices need calibration. Intrinsic calibration corrects the orthogonal relationship between the two laser emitters, allowing the simulator to generate crosshairs with coplanar and mutually perpendicular centerlines, thereby iso-centralizing the SCS with the RICS. The purpose of extrinsic calibration is to locate the simulator’s central position of RICS within the VCS (see Figure 3).

Although RICS can be vividly understood as three orthogonal planes, materializing them into a calibration model might hinder the visualization of laser projection due to occlusion, making it inappropriate for the crosshair simulator’s calibration. To address the issue, the present study introduces the concept of calibration spheres, which intuitively and visually express the geometric properties of RICS and solve the problem of light obstruction through spherical projection (see Figure 3 and Figure 4A).

Calibration spheres include both physical and virtual calibration spheres. The physical calibration sphere is made of solid, durable plastic with a radius of 60 mm. The scale lines on the calibration sphere include three bold, great circle arcs (GCA) of different colors (red, blue, black) and three dotted small circle arcs (SCA) (see Figure 4A). The intersections of the sphere define the three GCAs with three orthogonal great-circle sections passing through the center of the sphere, analogous to the earth’s equator, 0° + 180° meridians, and 90° E + 90° W meridians in geography. The three great-circle sections can be regarded as the three reference planes of the RICS. The three SCAs are defined by intersections with three equal small circle sections parallel to the corresponding colored great-circle sections. It is easily inferred from solid geometry that the three SCAs are tangent to each other on the sphere’s surface. Clearly, the planes containing arcs of different colors are necessarily perpendicular, and any combination of red-blue-black planes is mutually orthogonal. Hence, the three GCAs and three SCAs can form eight orthogonal combinations (see Figure 4C). Without loss of generality, the three GCAs define the unique primary calibration position (PCP) of the physical calibration sphere. In contrast, combinations containing SCAs define seven secondary calibration positions (SCPs) of the physical calibration sphere.

The virtual calibration sphere is a simplified form of the physical one, existing as a hologram in virtual space. It retains the three GCAs distinguished by red, green, and blue, strengthening the user’s understanding of this spatial relationship through some auxiliary lines within the red great-circle section. Since the SCAs are omitted, the virtual calibration sphere only contains the PCP.

The calibration process begins with intrinsic calibration, achieved through a physical calibration sphere. The simulator’s lasers can produce three projection arcs on the sphere’s surface. If the centerlines of the two sets of crosshairs are coplanar and mutually perpendicular, all three projected arcs should perfectly align with the marked arcs at one PCP and seven SCPs. Conversely, lasers that fail to maintain an orthogonal relationship will alter the curvature radii of some projected arcs, resulting in imperfect alignment between the projected and marked arcs across all calibration positions. During the intrinsic calibration process, the user first secures the calibration sphere at the PCP and makes fine adjustments to the lasers via screws to align the projected and marked arcs. Then the seven SCPs verify the calibration (see Figure 3 and Figure 4A–C). If satisfactory, the calibration sphere is returned to the PCP for extrinsic calibration.

The extrinsic calibration involves loading the virtual calibration sphere on the developed MR platform’s “calibration” panel (see Section 2.2.2.). The user can achieve precise translation or rotation adjustments to the virtual calibration sphere through the panel’s sliders, ultimately perfectly aligning it with the physical calibration sphere secured at the PCP (see Figure 3 and Figure 4D,E, and Appendix A).

Intrinsic and extrinsic calibrations will be only performed once, as the transformation matrices can be saved for further use in each subsequent registration.

#### 2.1.4. Mathematical Model for Crosshair Simulator-Based Registration

The mathematical model for crosshair simulator-based registration is shown in Figure 5. The key to crosshair simulator-based registration is to find the transformation $T_{W}^{Sc}$ between the scanner coordinate system and the world coordinate system defined by the HMD.

As mentioned, the two coordinate systems are in different spatiotemporal domains, so scanning parameters must be transferred through the crosshair simulator. In this way, $T_{W}^{Sc}$ can be calculated using the product of $T_{S}^{Sc}$ and $T_{W}^{S}$ as Equation (2):(2)$$T_{W}^{Sc}={T_{S}^{Sc}\cdot T}_{W}^{S}$$

Thereby, $T_{S}^{Sc}$ represents the transformation from SCS to the scanner coordinate system, determined by aligning the projected crosshair with the laser positioning lines marked on the patient’s head during image acquisition. $T_{W}^{S}$ represents the transformation from the world coordinate system to the SCS, which can be calculated as the product of $T_{V}^{S}$ and $T_{W}^{V}$ as Equation (3):(3)$$T_{W}^{S}={T_{V}^{S}\cdot T}_{W}^{V}$$

$T_{V}^{S}$ of the VCS to the SCS is determined during the calibration of the crosshair simulator. $T_{W}^{V}$ represents the transformation from the world coordinate system to the VCS, which can be calculated as Equation (4):(4)$$T_{W}^{V}={T_{H}^{V}\cdot T}_{W}^{H}$$
where $T_{H}^{V}$ is provided by HMD’s detection and tracking of the image target and $T_{W}^{H}$ is determined through the SLAM algorithm. By combining the above Equations (1)–(4), Equation (5) can be obtained:(5)$$T_{W}^{Sc}={T_{S}^{Sc}\cdot T}_{W}^{S}=T_{S}^{Sc}\cdot \left({T_{V}^{S}\cdot T}_{W}^{V}\right)=T_{S}^{Sc}\cdot \left[\left(T_{R}^{S}\cdot T_{V}^{R}\right)\cdot \left({T_{H}^{V}\cdot T}_{W}^{H}\right)\right]$$

Using $T_{W}^{Sc}$ as shown in Equation (5), the planned holographic model (based on the reference image or another image already registered to the reference image) can be transformed from its original position to the patient’s surgical site, enabling MRN.

### 2.2. The Components of the MR Platform

This subsection describes the hardware and software components of the MR surgical platform deployed on the HMD.

#### 2.2.1. MR HMD

The overarching goal of MR HMD is to understand the geometric transformation from the device to the environment and to display the holograms of virtual objects to users, effectively enabling the conversion and integration of virtual and actual entities. This study realized this goal using the commercially available Microsoft HoloLens-2 (HL-2) (Microsoft, Redmond, WA, USA) as the hardware component.

The HL-2 is an untethered, portable stereoscopic optical head-mounted computer boasting a small, lightweight independent computing unit (weighing 579 g) that operates without reliance on additional hosts or tracking hardware. It retails at approximately 1% cost of a standard commercial navigation system. Equipped with an integrated high-definition RGB camera for photography, four visible light cameras (VLC) positioned at different angles, a depth camera, and an inertial measurement unit (IMU), the HL-2 continually understands its external environment and pose (position and orientation) based on the flow of real-world information from the physical sensors and cameras. This mechanism enables the determination of the real-time geometric transformation between the HL-2, and the world coordinate system, embodying the principle of SLAM employed by the HL-2.

Further, through matrix multiplication, any object within the world coordinate system with a known or computable pose can establish a geometric relationship with the HL-2, culminating in the registration of holograms. The HL-2 offers a color display with a 43° × 29° field of view (FoV), respectively. By utilizing advanced waveguide imaging technology, the optical focus of virtual objects is fixed on a specific plane by the HL-2, thereby presenting them at their anticipated locations on the display screen. In essence, the integration of conversion and fusion between the virtual and actual on the HL-2 allows the operator to experience a stable, aligned position of the virtual object in the actual scene despite changes in viewing angles, thus offering a realistic visual experience.

#### 2.2.2. MR Platform Development

The MR platform is developed using Unity (Version 2021.3.4f1, Unity, San Francisco, CA, USA). The detection and tracking algorithm for the identification images is implemented based on the Vuforia SDK (Version 10.15). The interactive user interface uses the Mixed Reality Toolkit (MRTK) SDK (Version 2.8.3, Microsoft). Programming used C# scripts in Visual Studio (Version 16.11.26, Microsoft, 2019). The scripts handle the basic logic of the program and provide user-friendly voice and gesture commands. Finally, the project is packaged and deployed to HL-2 as a Universal Windows Platform (UWP) application.

### 2.3. Practical Workflow of MRN System

This subsection outlines the practical workflow of the MRN system, as illustrated in Figure 6. The principal steps include (1) laser projection marking, (2) image segmentation and hologram generation, (3) deployment of the crosshair simulator, and (4) hologram registration and updating.

#### 2.3.1. Image Acquisition and Laser Projection Marking

At the onset of the image acquisition, the user marks the laser positioning lines projected by the CT/MRI scanner on the patient’s skin. These marked lines will guide the correct deployment of the crosshair simulator in subsequent processes. Following this, a 3D imaging scan is conducted, and the obtained imaging data (i.e., the reference image) is exported from the workstation in Digital Imaging and Communications in Medicine (DICOM) format.

#### 2.3.2. Image Segmentation and Holograms Generation

The obtained 3D imaging data is imported into the open-source software 3D Slicer (Version 5.1.0). If other preoperative 3D imaging data are available for the patient, they are imported and fused with the reference images used for the registration process. Subsequently, surgical planning is undertaken, manually or semi-automatically, segmenting all surgery-relevant structures of interest, such as the skull, lesions, vessels, customized annotations, etc. Within the “Segment Editor” module of 3D Slicer, users can manually extract masks using essential tools like the paintbrush and eraser or semi-automatically obtain masks with an extended toolkit featuring intensity thresholds, islands, scissors, hollowing, smoothing, holes-filling, and logical operations. These techniques aim to minimize creation time and enhance segmentation quality. Lastly, the segmented structures are reconstructed into 3D virtual objects and exported in HL-2-compatible “.obj” format, so the holograms for MRN visualization are generated.

#### 2.3.3. Crosshair Simulator Deployment

The deployment of the crosshair simulator starts with the fixation of the patient’s head. The user activates the laser emitters and adjusts the simulator’s position until the projected lines perfectly align with the laser positioning lines previously marked on the skin of the patient’s head (see Figure 6 and Figure 7E). After that, the simulator’s pose is locked to maintain its relative position to the patient’s head.

#### 2.3.4. Holograms Registration and Update

Then, the user comfortably wears the HL-2 to prevent unexpected displacement. The MR platform utilizes Vuforia’s proprietary feature detection algorithm and the known image targets on the MR interface to achieve real-time tracking of the crosshair simulator’s position, enabling visualization of RICS. Subsequently, the holographic model is imported into RICS through gesture commands, initiating the initial registration for MRN.

Afterward, the user can choose to retain or update the registration result. This can be achieved by toggling the “Freeze” and “Unfreeze” states in the MR platform. “Freeze” locks the holographic model to the spatial anchor in the spatial mapping through SLAM of the HL-2, providing a more stable but static holographic perception even if the image target temporarily loses its tracking (see Figure 5 and Figure 7F). On the other hand, “Unfreeze” reactivates tracking of the image targets, enabling hologram updates in case of any changes in the patient’s head position in relation to the world coordinate system.

### 2.4. Experimental Design for Proof-of-Concept

#### 2.4.1. Image Data Source

To evaluate the feasibility of the proposed registration method, a 3D-printed head phantom based on a patient’s CT data was used (see Figure 7A,C). The imaging data was obtained from a 63-year-old male patient who underwent treatment at the Chinese PLA General Hospital in January 2021. The patient had a right-sided basal ganglia hematoma, and six CT-visible fiducial markers were attached to the scalp before the scan (Figure 7A). The CT scans were performed using a 128-slice CT scanner (Siemens SOMATOM, Forchheim, Germany) with the following parameters: tube voltage 120 kV, window width 120, window level 40, matrix size 128 × 128, FOV 251 × 251 mm^2^, and slice thickness 0.625 mm resulting in a voxel size of 1.96 × 1.96 × 0.625 mm^3^.

Before data acquisition, written informed consent was obtained from the patient’s authorized representative to use the pseudonymized imaging data for research purposes. Since no invasive procedures on any patient were involved in the study, no ethical review procedure was required.

#### 2.4.2. Head Phantom Creation

A 3D head model was created through semi-automatic segmentation using open-source software 3D Slicer (for further details, see Appendix A). The scan reference plane positions provided by the imaging data were used to restore the projection of the laser positioning crosshairs, which were then added to the model (see Figure 7B,C). The model was saved in “.stl” format and 3D printed using a commercial 3D printer, A5S (Shenzhen Aurora Technology Co., Ltd., Shenzhen, China), with the following parameters: nozzle temperature: 210 °C, platform temperature: 50 °C, material: polylactic acid, resolution: 0.3 mm, fill level: 10%, to create a 1:1 scale head phantom for evaluation purposes.

#### 2.4.3. Creation of Holograms for Validation

To fully validate the accuracy and reliability of the MRN system, a set of holograms was created using the 3D Slicer software for the experiment (see Figure 7D, for more details, see Appendix A). This included (1) the hematoma, (2) a puncture path, (3) fiducial markers, and (4) two quadrants of the scalp divided by the reference plane. The hematoma and puncture path were used to test the MRN system’s capability in handling and rendering simple or complex models. This is because, in practical applications, the number of triangular meshes composing a hologram can range from tens to thousands, providing information on geometric details and surface contours. The fiducial markers were used to quantitatively measure the TRE at specific points, as described in Section 2.4.5. The scalp quadrants provided the user with an intuitive impression of the overall alignment quality.

#### 2.4.4. MRN Registration and Holograms Deployment

Before deploying the holograms, the HL-2 was calibrated for the user’s interpupillary distance (IPD) to ensure an appropriate immersive visual experience. Then, the user’s planned holograms were registered to the fixed head phantom through the MR platform, as described above in Section 2.3.4 (See Figure 7E,F).

#### 2.4.5. Accuracy Evaluation

Once the hologram deployment was completed, the TRE was immediately analyzed to characterize the initial registration accuracy of the system. The TRE represents the deviation between the holographic representation of the fiducial markers in the virtual space and the corresponding fiducial markers in the physical space, not used for the registration process after performing the registration [38] and was calculated as the root mean square error (RMSE) of point pairs in three principal axes directions in virtual or physical space.

Six fiducial markers $\left\{A,\;B,\;C,\;D,\;E,\;F\right\}$ attached to the scalp were selected as the known reference points $P_{i}$ for measurement, as they were not involved in the registration process (see Figure 7G).

For each reference point, the user carefully and accurately placed the tip of the virtual probe on the perceived real-world marker point using their previous stereoscopic visual experience on the MR platform (see Figure 7H). The platform immediately reported the three-dimensional coordinates $Q_{i}$ of the probe tip in the RICS to the user’s panel. Then, the three-dimensional Euclidean distance ${\left\|P_{i}Q_{i}\right\|}_{2}$ between $P_{i}$ and $Q_{i}$ in the RICS was computed as Equation (6):(6)$${\left\|P_{i}Q_{i}\right\|}_{2}=\sqrt{{\left(P_{ix}-Q_{ix}\right)}^{2}+{\left(P_{iy}-Q_{iy}\right)}^{2}+{\left(P_{iz}-Q_{iz}\right)}^{2}}$$

Finally, three registrations were performed by a single user (Z.Q.). After each registration, the system switched to the “Freeze” state, and all six landmarks were sequentially targeted with the virtual probe. The PV camera then captured the panel display of the three-dimensional coordinate list, which was considered the result of one measurement session. Subsequently, after updating the holograms through the “Unfreeze” command, the system switched back to the “Freeze” state to conduct the next measurement session. Each registration included three rounds of measurement, thereof a total of 6 × 3 × 3 = 54 points that were acquired and used for TRE analysis.

To better visualize the TRE, the error distribution across the entire head was extrapolated for each measurement based on the deviations at the six reference points (see Figure 8C). This was achieved by determining the optimal transformation $T_{P}^{Q}$ from $\left\{P_{A},\;P_{B},\;P_{C},\;P_{D},\;P_{E},\;P_{F}\right\}$ to $\left\{Q_{A},\;Q_{B},\;Q_{C},\;Q_{D},\;Q_{E},\;Q_{F}\right\}$ using the least squares method. The original head model (P) was then transformed by this optimal transformation to produce the registered head model (Q). Finally, the “Model to Model Distance” module computed the absolute distance point by point between P and Q. The calculated distances were then mapped to Q as scalar values using color mapping.

#### 2.4.6. Statistical Analysis

Statistical analysis was conducted using one-way analysis of variance (ANOVA) to test the differences between the TREs of the three registrations and the six fiducial markers. The statistical significance level was set at *p* < 0.05. All statistical analyses were performed using MATLAB software (version R2022a, MathWorks, Apple Hill Campus, Natick, MA, USA).

## 3. Results

### 3.1. Workflow Analysis

The registration procedure was successfully implemented. Setting up the crosshair simulator position took about 2 to 3 min. Importing the holograms to the MR platform took about 1 to 2 min. For a detailed view of the user experience, a video is provided as Appendix A for an immersive holographic experience from the user’s perspective.

### 3.2. Accuracy Analysis

The average TRE across all points and registrations was 3.7 ± 1.7 mm, ranging between 1.2 mm and 8.9 mm (see Figure 8A–D). There, 81.5% of the measured points exhibited a TRE below 5 mm (see Figure 8A), a cut-off value established based on the previous study that developed and tested a fiducial-based MRN system for accuracy assessment [9]. There was no significant difference between the TREs between the three registrations (4.1 ± 1.9 mm, 3.3 ± 1.4 mm, and 3.6 ± 1.7 mm, *p* = 0.755), showing a high reproducibility of registration accuracy, nor was there a significant difference between the TREs of the six fiducial markers across all three registrations (3.5 ± 1.6 mm, 3.4 ± 1.6 mm, 4.0 ± 1.0 mm, 4.2 ± 2.2 mm, 3.2 ± 1.2 mm, and 3.7 ± 2.3 mm, *p* = 0.992) (see Table 1, Figure 8E).

## 4. Discussion

This study presents and assesses a novel MRN registration method utilizing a laser crosshair simulator to mitigate prevalent challenges concerning user dependency, cost-effectiveness, and accuracy in neurosurgical interventions. The developed system simulates the scanner frame’s position on the patient and autonomously computes the transformation, mapping coordinates from the tracking space to the reference image space. Preliminary evaluations using a head phantom indicated promising results, setting a foundation for future improvements and potential applications in clinical settings.

The pivotal role of MRN technology in neurosurgical interventions lies in its ability to offer clinicians an immersive environment. Deep-seated anatomical structures and planned surgical interventions can be visualized within this environment through the patient’s surface. Consequently, much research has confirmed the benefits of MRN, such as portable, low-cost, easy-to-use, free-hand interaction, intuitive understanding, and improved ergonomics compared to pointer-based and microscope-based conventional navigation systems [6,7,8,9,44,48]. The former requires neurosurgeons to continuously switch between surgical instruments and the pointer for navigation support, interrupting surgical operations and leading to potential distractions and fatigue. While offering advantages like superimposing virtual anatomical structures onto the surgical area, the latter comes with high costs and requires substantial hardware components. However, concerns about the accuracy and reliability of MRN were frequently raised on the other hand [9,10,30,37,39,41,42,43,49,50].

Ensuring a compelling holographic visualization experience for users is crucial, primarily achieved through the accurate overlay of virtual content onto the surgical field [30,34,38]. Therefore, in most publications concerning MRN-assisted surgical interventions, the emphasis was placed on registration, tracking, and technical challenges encountered.

The prevalent strategy for facilitating MRN support compels users to manually align the virtual objects with the patient’s physical counterparts [6,7,8,30,32,45,48]. From the user’s perspective, the virtual objects’ position, orientation, and scale are manually adjusted to align with the actual ones. Subsequently, through the HL-2′s inherent spatial map, the virtual objects are anchored to the patient’s actual location. This process negates the need for additional software or algorithmic resources. Nonetheless, this approach only yields statically registered holograms, necessitating a re-registration every time a spatial relation shifts between the patient’s head and the reference spatial map [9,47]. Moreover, the steep learning curve associated with manual registration implies it is both time-consuming and less dependable [6,7,8,9,43].

Landmark-based registration uses uniquely identifiable natural or artificial landmarks for paired-point matching. These markers can be pasted or anchored on the patient’s scalp or custom pointers which function in a similar way to the tracking pointer utilized in conventional navigation systems [9,10,44,49,53]. It is quicker to implement than manual alignment [9,38]. However, pre-shift of the markers or wear and tear of the pointers over time introduce fiducial localization errors (FLE) that impact the final registration accuracy. Notably, this method’s robustness depends on several factors, including the potential skin shift during imaging, the patient’s positioning in the OR, and the targeting of the marker with a physical pointer more robust [9,10].

Within the context of MRN, markerless registration, typically surface-based, distinguishes itself from manual or fiducial-based registration. Through computer vision (CV) algorithms, they automatically acquire partial surface information from the patient and correlate it with surface data from corresponding positions in the reference image, thus eliminating the time-consuming nature of complicated operations and the potential hazards of contact [33,54,55,56,57,58]. Nonetheless, the challenges of achieving robust and precise registration are magnified by the roughness of spatial mapping, and also the lack of distinctive feature points (e.g., in a prone registration session) and sensitivity to noise (e.g., geometric distortions in the original image, notably in an MRI image). Additionally, real-time tracking and rendering are necessary to enhance the signal-to-noise ratio of the captured surface data, demanding considerable computational resources [54,55,56]. Therefore, users may need to reduce the visualization expenditures, for example, by lowering the frame rate or decreasing the surface sampling rate [57]. The feature of Holographic remoting introduced by Microsoft in 2020 enabled the HL-2 to offload the rendering process to a remote computer or server and stream the rendered content back to the device for display. While this feature allows for leveraging more robust computing resources and alleviates rendering pressure on the HL-2, it also introduces complexity by making the system dependent on external computing resources, potentially increasing costs.

Although a vast amount of research has evaluated the accuracy of MRN systems based on various registration methods across various clinical intervention measures, novel registration methods that overcome these challenges still need to be developed. Therefore, this paper proposes the concept of simulator-based registration rather than focusing on specific neurosurgical intervention procedures.

Simulator-based registration offers a more straightforward and confidence-enhancing approach. During the registration process, the user’s task is merely to physically align two laser cross-projections, which is easier to achieve than touchless virtual object handling and eliminates the issue of pointer wear with no need to segment or predefine additional virtual objects for registration purposes (e.g., selecting virtual marker points or defining regions of interest for registration) as the scanned reference plane is already prepared and independent of the user in the reference image DICOM. These three reference planes are mutually orthogonal, imbuing the registration process with globally averaged characteristics. Furthermore, the system is easy to assemble and configure. On the hardware side, the production of the crosshair simulator is simple and comes with a low manufacturing cost. On the software side, the involved tracking and rendering do not require significant computing resources.

In the novel MRN system proposed in this study, the fusion of the crosshair simulator with the MR Platform offers valuable navigational advantages (see Table 2), which can be delineated in three key aspects:From a technical perspective, the crosshair simulator provides surgeons with an intuitive reference for physical positioning, while the MR Platform furnishes a visual reference for anatomical structures. This combination ensures reliable physical positioning and aids in a deeper comprehension of the surgical area.Regarding visual tracking, the crosshair simulator furnishes a stable CV tracking reference in physical space, whereas the MR Platform ensures visual stability through spatial anchors as the user moves. The spatial anchors signify holographic visualization optimization when surgeons need to relocate or change angles during surgery.In the context of practical workflow, the crosshair simulator can be rapidly deployed during surgical preparation, followed by the activation of the MR Platform to provide clear visual references and planning for surgeons. In the event of technical failures in either system, the Crosshair simulator can act as a backup physical reference location, while the MR Platform can concurrently optimize CV tracks. Hence, this fusion enhances efficiency, reliability, and robustness.

It is worth noting that fusion of the crosshair simulator and the MR Platform follows a structured approach, commonly referred to as “hybrid combination” in the mechatronics field. This concept, initially introduced by Zhang et al. [68], adheres to the principles of complementarity and compatibility. In essence, the complementarity principle implies that each technology compensates for the deficiencies of the other, while the compatibility principle ensures seamless integration of the two technologies [68]. Of more significance, this study introduces a novel registration method centered around the development and utilization of the crosshair simulator. The key innovation in this study revolves around applying the crosshair simulator to establish an identical coordinate origin as that of the CT or MRI. This foundational technique serves as the bedrock of the entire system, facilitating seamless integration and precise alignment, effectively addressing a critical challenge in MRN. While prior MRN studies were acknowledged, the specific contribution of this study primarily focuses on the comprehensive exploration and validation of the crosshair simulator’s pivotal role in achieving this coordination.

The crosshair simulator utilizes laser-based positioning. Although using lasers for preoperative or intraoperative localization is not a unique technique, such as obtaining surface data of patients [5,57,69,70], indicating planned puncture approaches [71,72], or guiding the correct positioning for radiotherapy [45,73], they typically provide point information from laser projections. However, the principle employed in this study entirely differs from previous applications, as it leverages both point and normal vector information in the projection transformation of the laser crosshair on 3D surfaces.

In the presented approach, a pair of laser crosshairs was used, where the projection of a single crosshair on a 3D curved surface can constrain two translation degrees of freedom (DOFs) (excluding the normal vector direction) and three rotational DOFs for the simulator. Another crosshair’s projection direction is perpendicular to the first, compensating for the missing translational DOF and providing redundant constraints to ensure reliability and accuracy. Additionally, the rigidly fixed relationship between the laser emitters constitutes a constraint on the scale factor. Therefore, through a complete constraint of six DOFs plus scale, a one-to-one mapping between the crosshair simulator and the projected lines on the patient’s head is achieved, enabling the simulator to reproduce its position during the acquisition of reference images by the scanner. The calibration of the simulator’s intrinsic and extrinsic parameters using calibration spheres is similarly based on this principle.

Although the proposed method shares some similarities with the standard optical navigation paradigm employing surface matching techniques based on a non-contact laser pointer, such as exploring the patient’s surface structure data through laser projection, it also has key differences. In the data acquisition stage of standard optical navigation, the user must manipulate a laser pointer (e.g., Z-Touch) to ‘capture’ a relatively non-large number (around 100–200) of random points from the patient’s craniofacial area, allowing for interpolation of the local shape of the surface. Nonetheless, in the proposed framework, the key lies in how the laser crosshair lines ‘twist’ on the surface. Once these projection lines perfectly match the reference lines, it effectively determines the position of the reference plane in image space. In this process, the curvature of the surface plays a critical role. Theoretically, the curvature of 3D surfaces may affect the efficiency of DOFs constraints. It was found that regions with smaller facial curvature radii, such as the nasal and zygomatic regions, have more significant surface curvature and higher variations in normal vectors. Even slight deviations in the laser projection angle from the ideal position on the 3D surface can cause significant deformation in the crosshair projection, leading to a mismatch with the reference lines (see Figure 9B). Therefore, compared to flatter and smoother regions with larger curvature radii, such as the forehead, the non-flat regions have higher DOFs constraint efficiency. They are, therefore, more critical in the registration value (see Figure 9A,B).

Although the term “accuracy” of MRN systems is not consistently defined, the TRE is a commonly used metric for evaluating navigation accuracy at different stages of the surgical intervention, ranging from initial registration [38] to registration quality in the late course of surgery. While other issues might arise later in the surgical procedure, such as non-linear deformations caused by brain-shift, the importance of initial registration should not be overlooked. The accuracy of preliminary registration will directly impact the accuracy and reliability of subsequent steps. TRE is defined as the distance between specifically selected points in virtual space and their corresponding points in physical space [34]. In this study, the TRE of the MRN system based on the crosshair simulator was measured in 3D space, with an average of 3.7 ± 1.7 mm and a range between 1.2 mm and 8.9 mm. Therefore, its accuracy may not be sufficient for neurosurgical microsurgeries requiring high accuracy yet but is acceptable for non-microsurgeries (e.g., extra-ventricular drainage (EVD)) and the macroscopic parts of microsurgeries (e.g., craniotomy planning). The noted accuracy is also comparable to that in fiducial-based registration. The TRE as presented by the system represents the deviation at the surface level. Although using markers like fiducials or bone screws allows for the acquisition of points both externally and within the images to calculate the Euclidean distance, this measure still does not reflect the actual target level since the target (e.g., tumor) is inaccessible at that stage. Recognizing this limitation, which is also observed in other studies, alternative non-invasive methods might be explored in future research. For instance, using non-invasive markers or imaging techniques to indirectly estimate the TRE at the deeper target level. However, it is acknowledged that directly assessing the target, especially if it is a tumor, remains challenging at this stage. A literature review on 3D TRE of MRN systems implemented on standalone HMDs is presented in Table 3 [7,8,32,33,35,36,44,48,50,51,52,56,58]. However, it must be emphasized that these studies cannot directly compare TRE due to their different purposes or measurement methods [38].

Previous work has categorized the measurement methods of 3D TRE into two types: direct and indirect. Calipers or commercial navigation probes are typically used to directly read the error in the former way [32,33,35,36,50,51,52,56,58]. It is worth mentioning that a recent study reported a more precise and efficient measurement by cleverly utilizing customized probes within the MRN system [49]. Nonetheless, the accessibility of the measurement tools could lead to an underestimation of TRE, as virtual targets may be located beneath the patient or phantom. In indirect measurements, users usually place a puncture needle or other markers at the position of the virtual target to mark its location [7,8,44,48,58]. Then, the position is re-registered onto the original image through “post-operative” imaging to calculate the re-registration error as TRE. Nevertheless, this measurement method may introduce changes to the original structure due to needle insertion (e.g., the shrink of the ventricle after EVD), potentially affecting the measurement’s repeatability and reliability [7,8]. Therefore, in this study, a direct and non-invasive approach was adopted where the relative point of the actual marker in the virtual space and measured the distance to the corresponding point of the virtual target was manually obtained. Placing the virtual probe at the same marker point multiple times in each registration helps control random errors caused by the lack of tactile feedback. However, a potential system error may be introduced because a 3D-printed head phantom was used instead of an actual patient’s head. Although using 3D-printed phantoms as ground truth is a common practice for preclinical validation of new systems’ feasibility, it may neglect various aspects that could occur during an actual acquisition. This includes skin shift when the patient’s head is fixed in the head clamp, deformations caused by intubation or the insertion of a nasogastric tube, unknown changes in muscle tension leading to skin deformation, and the effects of head positioning using headphones or foam pads for MRI. Additionally, it neglects the geometric distortions of the original imaging itself, such as those caused by resolution and slice thickness [74].

Nonetheless, the presented registration method does have some limitations. Firstly, restrictions on registration and tracking arise from software and hardware. The current system introduces a dedicated Vuforia SDK for real-time tracking through the PV camera of HL-2. While plane target deployment to the crosshair simulator is straightforward, as many studies have shown, the user’s perspective would affect the tracking quality of planar image targets [45,53,75]. When users observe from positions with large incident angles, the HL-2 tracking becomes unstable, leading to slight jitter or even loss of tracking. This impacts the accuracy of initial registration and affects the stability of the holographic content after registration. To overcome or limit this analysis, observing the holograms from angles perpendicular to the marker image yielded the most optimal visualization results. As for maintaining the holographic content after registration, a “Freeze” mode was developed that utilizes anchors in the HL-2′s spatial mapping to temporarily maintain the hologram’s position, replacing real-time image target tracking. This prevents inappropriate updates to the holographic content when users are in unfavorable angles. However, as reported in previous literature, a holographic drift was also observed in the “Freeze” mode. This drift may be due to the built-in sensors’ hardware limitations and the spatial mapping coarseness of HL-2. They are leading to an accumulated estimation error in the SLAM over time. Nevertheless, loop closure detection can eliminate this drift when switching to the “Unfreeze” mode, which re-enables real-time tracking of the image target. Nonetheless, whether the “Freeze/Unfreeze” switching can be reasonably triggered depends on the user monitoring whether the virtual content has deviated from the registered position. A potential improvement strategy could involve replacing plane targets with cylindrical surface targets to reduce user reliance on monitoring. In this scenario, the PV camera can detect the image target perpendicular to it at any angle, allowing for stable tracking and positioning.

Second, the potential challenges of using this novel registration method for surgery in the prone position should be considered. In most cases, preoperative CT or MRI scans are performed in the supine position, making the marking of projection lines for surgery in the prone position impractical. The importance of considering the surgical position in the preoperative imaging process is highlighted in recent studies. Furuse et al. evaluated the clinical accuracy of navigation systems [76]. They emphasized that the surface-based registration method was only recommended for the supine position due to skin distortion frequently observed in the lateral region. In contrast, Dho et al. demonstrated the superior accuracy of neuronavigational localization with prone-position MRI during prone-position surgery for posterior fossa (PF) lesions [77]. These findings suggest that acquiring preoperative data in the prone position, or using intraoperative imaging where a prone scan can be obtained, can improve registration accuracy. This would make registration based on the crosshair simulator feasible. Therefore, future research should broaden the applicability of the crosshair simulator for registering prone patients after an intra-operative scan or consider acquiring preoperative imaging in the prone position to align with the surgical position for more precise localization.

Thirdly, the calibration process of the crosshair simulator remains user-dependent. On the one hand, intrinsic calibration relies on the user’s assessment of whether the laser projection lines perfectly match the calibration arcs. Based on this, the coplanarity and verticality of the two laser emitters are adjusted. On the other hand, extrinsic calibration necessitates the user to manually align the virtual calibration sphere with the physical calibration sphere through slider adjustments on the MR platform. Errors generated by the user during both processes can affect calibration accuracy, subsequently impacting the overall performance of the MRN system. Future research endeavors should explore methods to eliminate this dependency. One potential improvement strategy for intrinsic calibration is to equip the physical calibration sphere with a self-changeable robot calibration module [78,79,80], effectively transforming it into a self-reconfigurable robot calibration sphere. Like the “self-docking” method reported by Wu et al. [80], this can be achieved by mounting laser sensors or cameras on the physical calibration sphere, allowing it to autonomously locate the target module on the rack and laser emitters, thus self-reconfiguring and establishing the PCP or SCP relationships. Regarding extrinsic calibration, equipping the physical calibration sphere with CV targets could enable automated geometric calibration. However, the challenge lies in the psychophysical aspect, where an aligned geometric position of the virtual calibration sphere does not guarantee a correct position within the user’s perceptual system due to the perspective difference between the HL-2’s PV camera and the user’s eyeballs. Lastly, this registration method only partially eliminates user dependencies. Although the marker projection lines were printed with the phantom in the experiments to simplify the procedure, the process becomes more complex in a clinical environment. During MRI or CT reference image acquisition, users may need to manually draw the marker lines on the patient’s skin at the beginning of the acquisition session. Therefore, readers may need clarification on whether these marking lines are robust and valid, especially in MRI data collection involving multiple localization procedures. In fact, an MRI scanning begins with a ‘localizer scan,’ defining the RICS at the position where the gantry laser positioning lines are projected. Subsequent localizers (defining the ROI for different modality scanning sequences) are set within this coordinate system without altering the system itself. Thus, any tilt or movement in later scans will not affect the validity of the marking lines. If the patient’s head position changes between different scans, images can be merged or fused using open-source software like 3D Slicer, ensuring that the marker lines remain valid. Although this might sound complex, the fusion of image series is a standard procedure in multi-modal navigation and can be easily implemented using 3D Slicer. So, the marking lines can ensure robustness and validity. Moreover, another reasonable concern is whether making markings on the skin (such as drawing lines) could introduce geometric distortions. While evidence from adjacent disciplines supports the routine use of therapeutic room lasers and skin marker lines for pre-treatment positioning [45,73], the specific impact of manual markings on the accuracy of registration based on the crosshair simulator still needs to be determined within the following context: the background of neurosurgery. Additional challenges that must be considered include calibrating the scanner’s tilted gantry, concerns about additional radiation, and challenges in users’ holographic perception. These aspects warrant further investigation and remain subjects for future research.

Besides those rather general limitations of the presented approach, there are also study-specific limitations. First, this proof-of-concept study only encompassed an experiment involving a single phantom by a single user, needing more reliability of multiple tests and statistical analysis. Furthermore, the current research framework does not enroll human participants, hindering a comprehensive assessment of its performance in clinical settings. To further explore its clinical adaptability, systematic testing on human volunteers is planned. For real patients, ensuring the stable fixation of the crosshair simulator to the head is paramount. Consideration can be given to using a rigid headframe connected to the patient’s bed or operating table. Medical tape or other fixation devices are also viable alternatives for ensuring the stability of head positioning during the surgical procedure. Given patient comfort and safety, these fixation solutions should be easy to rapidly install and dismantle. These strategies aim to enhance positional accuracy and reduce errors arising from head movement.

Despite the limitations, the novel registration method demonstrates promising prospects and merits further development. Compared to previous techniques, using laser crosshair alignment offers a more intuitive and efficient approach to registration, reducing the complexities associated with virtual object handling. The simplicity in hardware and software configuration, combined with the potential to enhance accuracy and adaptability in intervention procedures, positions this method as a valuable advancement in low-cost, easy-to-use MRN systems, potentially offering improvements in surgical outcomes.

## 5. Conclusions

This study presented a novel registration method for an MRN system based on a laser crosshair simulator. It provided initial evidence of its feasibility as a low user-dependent, cost-effective, and relatively reliable approach. The method showed encouraging results in efficiency and intuitiveness compared to previous techniques. Its simplicity in both hardware and software configuration, combined with the potential for enhancing accuracy and adaptability in intervention procedures, marks this method as a valuable advancement. Although refinements in accuracy are still possible, the current study lays the groundwork for improvements in low-cost, easy-to-use MRN systems, positioning this approach as a promising avenue for enhancing surgical outcomes. Future research may continue to build upon these strengths while evaluating their utility and effectiveness in broader clinical settings.

## Figures and Tables

**Figure 1 bioengineering-10-01290-f001:**
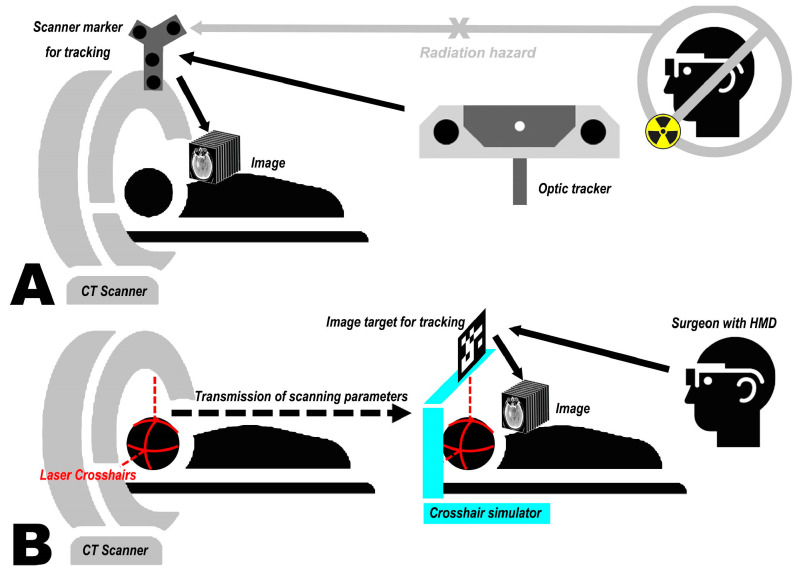
Mixed reality neuro-navigation (MRN) registration using scanner tracking (**A**) is challenging due to tracking signal interruption, as the user must stay away from the scanner during data acquisition. MRN registration using the proposed concept of a crosshair simulator (**B**) transferring the laser crosshair (red) location from the scanner to the simulator (cyan) enabling the projection of corresponding imaging data onto the patient’s head (HMD = head mounted display).

**Figure 2 bioengineering-10-01290-f002:**
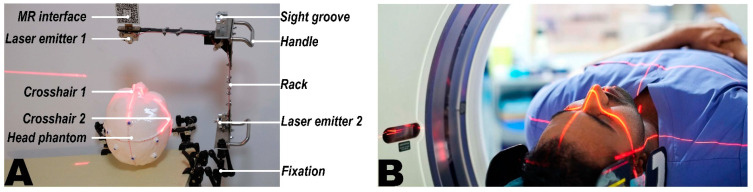
Structural and functional demonstration of the crosshair simulator. The crosshair simulator (**A**) exemplifies its ability to simulate scanner-generated laser crosshairs, forming two crosshair laser projections on a patient’s head, identical to those observed in CT or MRI scanners (**B**). Figure 2B provided by ©Alamy. The image has been granted copyright authorization from the Alamy platform.

**Figure 3 bioengineering-10-01290-f003:**
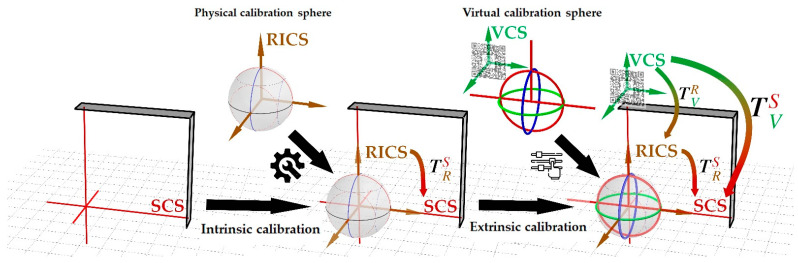
The three coordinate systems in the crosshair simulator (SCS = simulator coordinate system (red); RICS = reference image coordinate system (brown); VCS = virtual coordinate system (green); The transformations (colored arrows) from one to the other coordinate system are color-coded as gradients of the related coordinate systems.) and the mathematical expression utilizing both physical and virtual calibration spheres to calibrate the intrinsic and extrinsic matrices.

**Figure 4 bioengineering-10-01290-f004:**
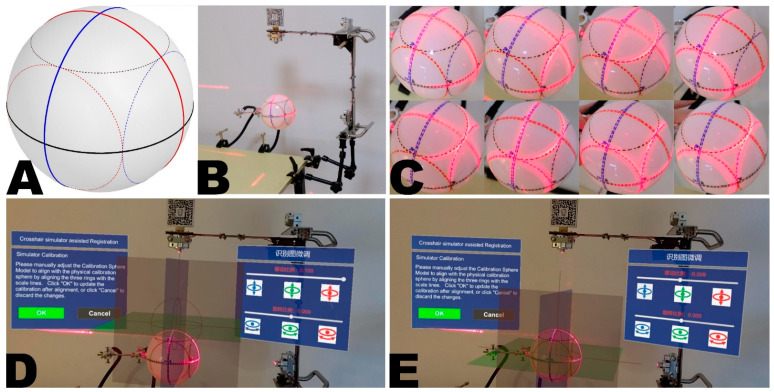
The calibration procedure of intrinsic and extrinsic matrices using a custom calibration sphere in the crosshair simulator. The structure of the calibration sphere is depicted in schematic form (**A**) and was used for calibrating the intrinsic of the crosshair simulator (**B**). During intrinsic calibration, eight known orthogonal calibration arc combinations on the sphere can be used (**C**). The process of extrinsic calibration is shown, with the virtual calibration sphere representing RICS not aligning with the physical calibration sphere (**D**). After adjustment, the virtual calibration sphere perfectly aligns with the physical calibration sphere, signifying successful calibration (**E**).

**Figure 5 bioengineering-10-01290-f005:**
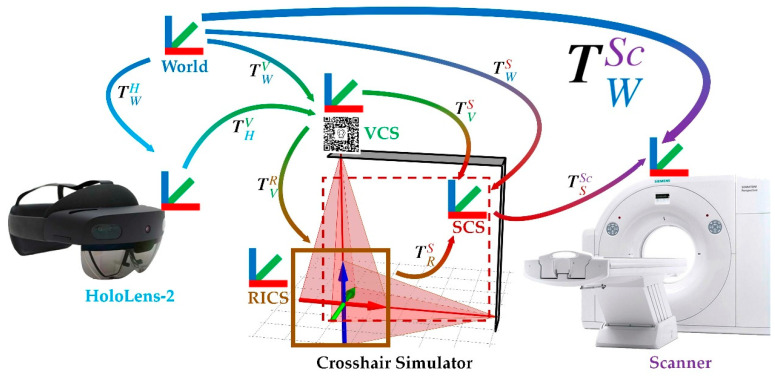
The mathematical model for crosshair simulator-based registration with coordinates systems (i.e., the world (blue),scanner (purple), RICS (brown), SCS (red), VCS (green), and HoloLens-2 (light blue) coordinate systems) and the related transformations between those (gradient color-coded).

**Figure 6 bioengineering-10-01290-f006:**
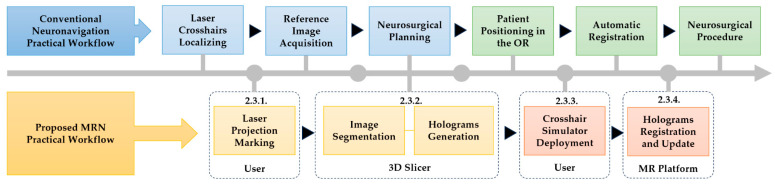
Practical workflow chart for the proposed MRN system compared with conventional neuronavigation system. (OR = operating room).

**Figure 7 bioengineering-10-01290-f007:**
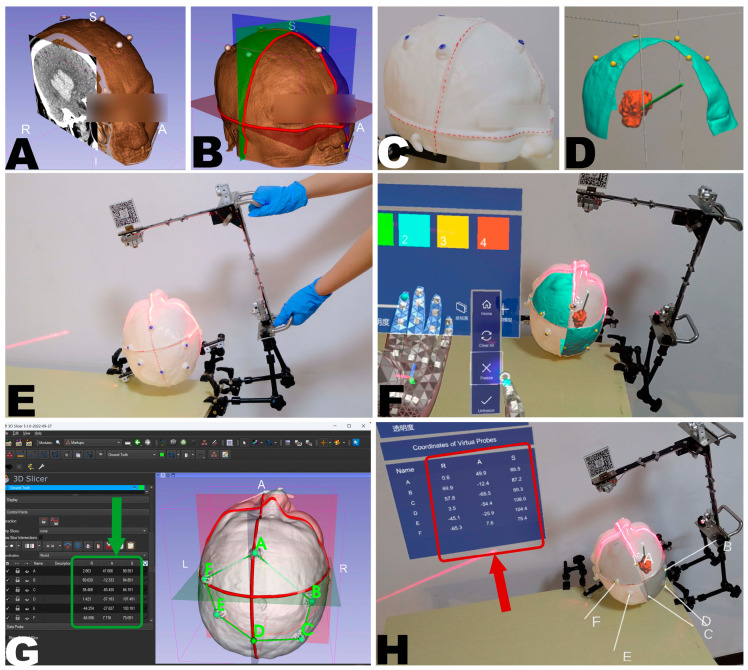
An illustrative case demonstrates the crosshair simulator-based registration process and its accuracy measurement method. CT imaging data, including visible fiducial markers attached to the scalp of a patient presenting with a right basal ganglia hematoma (**A**). The CT scanner’s laser crosshair projection lines on orthogonal reference planes were recreated using 3D Slicer (**B**). A 1:1 scale 3D printed model was generated with laser projection lines marked in red (**C**). Using 3D Slicer, a set of holograms for validation, including the hematoma in red, a puncture path in green, fiducial markers in yellow, and the two quadrants of the scalp divided by the reference plane in cyan, was created (**D**). Manual adjustment of the crosshair simulator showing a perfect match of the laser crosshairs and the marked laser positioning lines on the head model (**E**). Successful hologram registration perfectly aligned the holographic image with the 3D-printed head model (**F**). Coordinates of six fiducial points in the RICS, as shown in the green box, were selected for accuracy measurement using 3D Slicer software (**G**). Following MRN registration, the user positioned the virtual probe, consisting of a white line handle and a white spherical tip, on the perceived real-world fiducial points, as shown in the red box (**H**).

**Figure 8 bioengineering-10-01290-f008:**
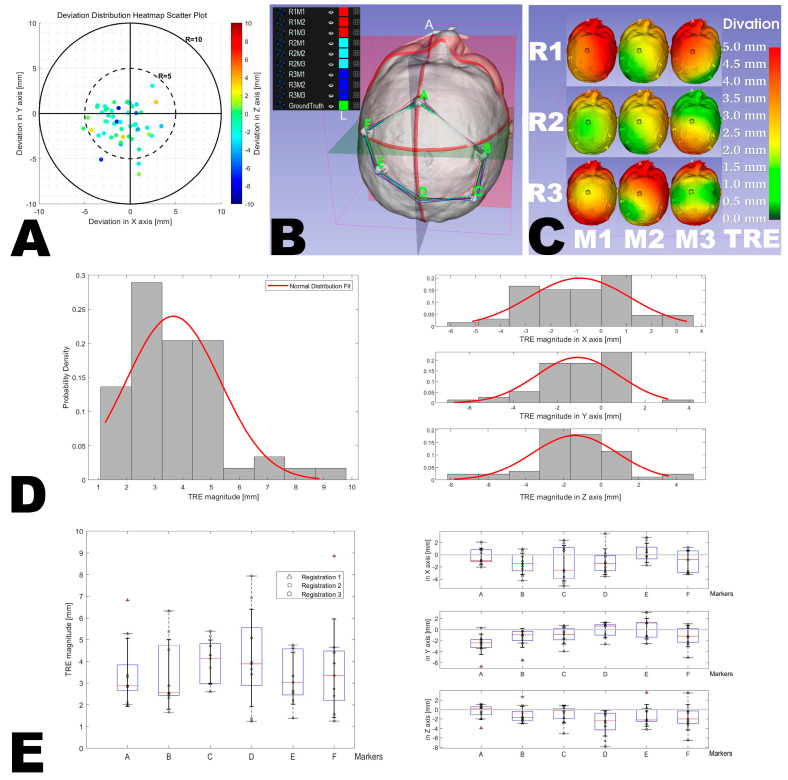
Results of accuracy measurement. A color gradient scatter plot demonstrates the deviation across all measured points (**A**). Overlapping polygons colored in red, cyan, and blue depict deviations for three different registrations on the original model (Legend: R = registration, M = measurement, and the numbers denote the respective sessions, e.g., ‘R1M1′ corresponds to the first registration followed by the first measurement) (**B**). The measured points extrapolated the full-head error distribution for nine sessions from R1M1 to R3M3 (**C**). The histograms present the distribution of deviations and their components along the X, Y, and Z axis (**D**). Box plots compare inter-group deviations grouped by fiducial points (Makers A, B, C, D, E, F) and deviations in the X, Y, and Z components (**E**). The whiskers represent the minimum and maximum values within 1.5 times the interquartile range (IQR) from the first (Q1) and third (Q3) quartiles. Any data points beyond this range, which are considered outliers, are marked with red crosses (+).

**Figure 9 bioengineering-10-01290-f009:**
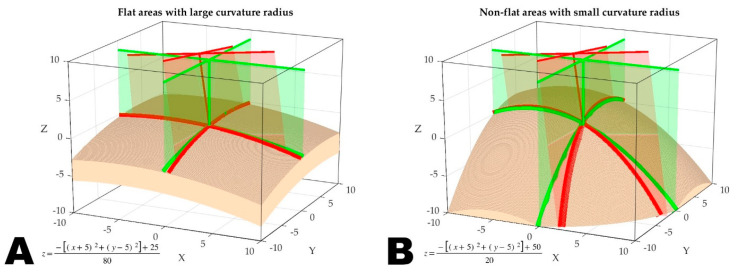
Comparison of laser crosshair projection on areas with large curvature radius (**A**) and small curvature radius (**B**). The projection simulation was conducted in MATLAB R2022a, and to simplify calculations, two parabolic surfaces with different apertures and curvatures were plotted. Assuming a slight angular disparity between the simulator laser (depicted as the red line) and the scanner laser (depicted as the green line) during the deployment of the crosshair simulator, rather than being perfectly coaxial, the projection of the simulator’s crosshair on the patient’s skin (depicted as the red curve) will distort at the reference lines (depicted as the green curve), resulting in an imperfect match. This mismatch is more pronounced in areas with a smaller curvature radius, aiding users in promptly detecting registration errors and adjusting the simulator’s deployment. Hence, these regions are more critical for the value of registration.

**Table 1 bioengineering-10-01290-t001:** Accuracy data for all registrations and markers.

Group	TRE * [mm]	Min [mm]	Max [mm]
Registration 1	4.1 ± 1.9	1.2	8.9
Registration 2	3.3 ± 1.4	1.4	6.9
Registration 3	3.6 ± 1.7	1.2	7.9
Marker A	3.5 ± 1.6	2.0	6.8
Marker B	3.4 ± 1.6	1.6	6.3
Marker C	4.0 ± 1.0	2.6	5.4
Marker D	4.2 ± 2.2	1.2	7.9
Marker E	3.2 ± 1.2	1.4	4.8
Marker F	3.7 ± 2.3	1.2	8.9
Total	3.7 ± 1.7	1.2	8.9

* Mean ± Standard deviation.

**Table 2 bioengineering-10-01290-t002:** Complementarity and compatibility analysis of the crosshair simulator and MR Platform.

Aspect	The Crosshair Simulator	The MR Platform	Complementarity	Compatibility
Technical principle	Provides physical positioning	Offers 3D visualization and virtual interaction	Combines benefits of physical and 3D visualization	Calibration process ensures synchronization and consistency between the crosshair simulator and MR
Visual tracking	Fixed visual tracking reference	Spatial anchors stabilize the MR view	Physical location back up and optimized CV vision tracking	Seamless transition between the two tracking modes
Practical workflow	Rapid physical positioning with low user dependency	Provides 3D visual references once activated	Quick re-registration; Remedial static guidance	Efficient and intuitive registration, simplifying interaction

**Table 3 bioengineering-10-01290-t003:** Literature Review of 3D TRE measured in Standalone HMD-based MRN Systems.

Reference	Registration Method	Object	Measurement Method *	Accuracy # [mm]
Li et al., 2018 [7]	Manual	Patient	Indirect	4.34 ± 1.63
Li et al., 2023 [8]	Manual	Patient	Indirect	5.46 ± 2.22
Gibby et al., 2019 [48]	Manual	Phantom	Indirect	2.50 ± 0.44
McJunkin et al., 2019 [32]	Manual	Phantom	Direct	5.76 ± 0.54
Zhou et al., 2022 [36]	Fiducial	Phantom and Patient	Direct	Phantom: 1.65 Patient: 1.94
Gibby et al., 2021 [44]	Fiducial	Phantom	Indirect	3.62 ± 1.71
Gsaxner et al., 2021 [51]	Fiducial	Phantom	Direct	1.70 ± 0.81
Martin-Gomez et al., 2023 [52]	Fiducial	Phantom	Direct	3.64 ± 1.47
Zhou et al., 2023 [35]	Fiducial	Phantom	Direct	1.74 ± 0.38
Eom et al., 2022 [50]	Fiducial	Phantom	Direct	3.12 ± 2.53
Akulauskas et al., 2023 [49]	Fiducial	Phantom	Direct	Stationary: 3.32 ± 0.02 Dynamic: 4.77 ± 0.97
Von Haxthausen et al., 2021 [56]	Surface	Phantom	Direct	14.0
Pepe et al., 2019 [33]	Surface	Phantom	Direct	X: 3.3 ± 2.3 Y: −4.5 ± 2.9 Z: −9.3 ± 6.1
Liebmann et al., 2019 [58]	Surface	Phantom	Indirect	2.77 ± 1.46

* Two types of measurement methods: direct and indirect. Direct: The distance between the real target and the perceived virtual target is directly measured by a caliper or a tracked probe linked to a reliable optic tracking system (e.g., conventional navigation). Indirect: TRE is characterized by the distance between the annotated target and the tip of a catheter or needle measured on the post-operative image. # Mean ± Standard deviation.

## Data Availability

The data presented in this study are available upon reasonable request from the corresponding authors.

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
