# Peer review of "A Novel Registration Method for a Mixed Reality Navigation System Based on a Laser Crosshair Simulator: A Technical Note"

_bioengineering, 2023, doi:10.3390/bioengineering10111290_

Round 1

Reviewer 1 Report (Previous Reviewer 2)

Comments and Suggestions for Authors

The paper is fine. No further comments.

Reviewer 2 Report (Previous Reviewer 3)

Comments and Suggestions for Authors

I am satisfied with the revision along with the rebuttal.

Comments on the Quality of English Language

good

This manuscript is a resubmission of an earlier submission. The following is a list of the peer review reports and author responses from that submission.

Round 1

Reviewer 1 Report

Comments and Suggestions for Authors

Thank you for inviting me to review this very interesting manuscript. 

I have read the manuscript with interest. It is well performed and well-balanced and considers the most important aspects of navigation and mixed reality. The method description is adequate and reproduciable. The limitations of the method have been identified and discussed. 

Although the accuracy achieved is still below requirements for precision surgery, the method offers a very good starting point for future development. The authors should be commended for their excellent work.

Comments on the Quality of English Language

-

Reviewer 2 Report

Comments and Suggestions for Authors

The study describes a new markerless registration system for neurosurgery based on a cross hair laser. This topic is a known problem in navigation for surgery.

The system is described properly and presented well. The authors describe the systems they are using and the experiment they've done to test their system.

The system is tested on a 3D printed model. Is it possible to test it on a volunteer?

A bit more details are required on the fixation of the real patient.

The English can be improved.

Comments on the Quality of English Language

Samples:

"The system autonomously 17 calculates the transformation from the tracking space to the reference image space" - please revise

"Neurosurgeons must switch surgical instruments with the pointer when nav-38 igation support is needed, thus interrupting surgical operations" - must?

...

Reviewer 3 Report

Comments and Suggestions for Authors

Comment 1:

The paper presents a novel registration method based on a laser crosshair simulator. However, the work appears to be published before by the team of the authors. The specific contribution of this paper is thus little.

Comment 2:

The idea of Mixed Reality Navigation (MRN) is to combine several approaches, but in the literature of combination, there is one called hybridization and thus hybridization engineering, see “A novel hybridization design principle for intelligent mechatronics systems”. The authors may want to elaborate on their specific combination, i.e., whether it is a hybrid combination. It is noted that the hybrid combination is the optimal one.

Comment 3:

The idea of the vision system combined with some depth sensor is not new in broader field of robotics, e.g., there is a work to use laser sensor with camera to perform automatic docking in robotics, see “J. Q. Wu, et al., "A Novel Self-Docking and Undocking Approach for Self-Changeable Robots," 2020 IEEE 4th Information Technology, Networking, Electronic and Automation Control Conference (ITNEC), Chongqing, China, 2020, pp. 689-693, doi: 10.1109/ITNEC48623.2020.9085076.” In this work, the robot is search an object and then dock with it. Although such an idea may be new in the field of image registraiton in the case of neurosurgey, as an academic paper, we need to act as general as possible to communicate with the reader about this information.

Comments on the Quality of English Language

good.

Reviewer 4 Report

Comments and Suggestions for Authors

The authors describe a registration method for a mixed reality system by means of a laser crosshair simulator and further evaluate it with a head phantom while measuring the target registration error. They use the simulator to find the same coordinate origin as that of the CT or MRI, which is the key aspect of this work. The manuscript is very detailed and well organized, allowing the reader to have a good understanding of this approach. 

Major points

  • In the discussion, the authors talk about user dependencies regarding the marker projection lines. In my opinion, the extrinsic calibration process also contains user dependency as the user needs to manually place the virtual calibration sphere on top of the real one. The authors should critically discuss to what extent this has an influence on the calibration result.

Minor points (lines)

62

HMD usually refers to head mounted display

89

There seems to be one "the" too much

207

Why is there an apostrophe at the transformation?

264 ff

It seems that extrinsic and instrinsic are mixed up in the caption

304

Space missing after and

316

Again an apostroph

460

It should be mentioned that the virtual probe consists of the white lines / pointers

Figure 8a

I assume that one axis should be the deviation in Y axis

Figure 8b

Please describe the legend entries in the caption

Figure 8e

Please describe how the whiskers are defined

427

Please mention the printing parameters used

Comments on the Quality of English Language

English language is mostly fine, just some minor mistakes.
